# Microwave-Field-Optimized GO/TiO_2_ Nanomaterials for Enhanced Interfacial Charge Transfer in Photocatalysis

**DOI:** 10.3390/nano14231912

**Published:** 2024-11-28

**Authors:** Xu Duan, Weizao Liu, Jing Guo

**Affiliations:** 1College of Materials Science and Engineering, Chongqing University, Chongqing 400044, China; sycamoreduan@cqu.edu.cn; 2School of Chemistry and Chemical Engineering, North University of China, Taiyuan 030051, China; 3Shanxi Province Key Laboratory of Chemical Process Intensification, North University of China, Taiyuan 030051, China

**Keywords:** microwave field, hydroxyl radical, narrowed bandgap, graphene oxide, TiO_2_

## Abstract

The swift recombination of photo-induced electrons and holes is a major obstacle to the catalytic efficiency of TiO_2_ nanomaterials, but the incorporation of graphene oxide and out-field modification is considered a potent method to augment photocatalytic properties. In this work, a series of GO/TiO_2_ photocatalysts were successfully optimized by a microwave field. As determined by transient photocurrent response and electrochemical impedance spectroscopy (EIS) tests, microwave irradiation at 600 W for 5 min on the GO/TiO_2_ photocatalyst promoted interfacial charge transfer and suppressed charge recombination. Through systematic characterizations, GT(600/5) exhibited the highest photooxidation rate (81.5%, 60 min) of Rhodamine B under visible light compared to other homologous samples, owing to the minimum grain size (16.914 nm), enlarged specific surface area (151 m^2^/g), maximum light response wavelength (510 nm), narrowest bandgap width (2.90 eV), and stronger oxidized hydroxyl radicals (•OH). Given the environmental friendliness, greenness, and sustainability, this study could present an efficient and economical strategy for synthesizing and fine-tuning photocatalysts.

## 1. Introduction

Over the decades, extensive photocatalytic semiconductor research has been conducted for environmental governance, including water purification, soil remediation, and flue gas removal [1,2,3,4]. Among these semiconductors, TiO_2_ has occupied a dominant position, owing to its nontoxicity, greenness, cheapness, and easy availability [5]. Still, the swift recombination of photo-induced electrons and holes is a major obstacle to the catalytic efficiency of TiO_2_ nanoparticles during practical applications [6]. Additionally, the effectiveness of degradation might be hindered by the inadequate absorption of UV light energy, as TiO_2_ is part of a class of wide-bandgap semiconductors. Researchers have conducted a lot of work to address the above two issues, mainly using doping, composite, and modification methods to modify their morphology and structure [7,8,9].

The discovery of graphene and graphene oxide has injected new blood into the field of photocatalysis [10]. Graphene is a zero-bandgap material that, when combined with semiconductor materials, can broaden the excitation wavelength range and improve the utilization efficiency of visible natural light [11]. GO, as the oxide of graphene, contains a large number of oxygen-containing functional groups on its surface, such as hydroxyl, carboxyl, epoxy, etc., [12]. These functional groups make GO more reactive than graphene and more likely to react with other substances. Photocatalysts based on graphene have been developed successively in laboratories [13,14,15,16], as GO exhibits good hydrophilicity and chemical reactivity. GO-TiO_2_ composite material (TGO composites) is regarded as an ideal photocatalyst, commonly prepared using conventional hydrothermal or solvothermal methods [17]. A few reports are now focusing on the merits of microwave-assisted hydrothermal processes to improve the crystallinity and uniformity of TGO composites within short timeframes [18,19,20]. However, there are still many problems that need to be solved to achieve large-scale industrial production outside of the laboratory.

During traditional photocatalyst preparation or modification, common heat sources for heating are used, which leads to problems such as uneven heating, long heating times, and temperature gradients [21,22]. Microwave treatment is a molecular-level modification technique that can directly act on molecules, intensifying their vibration and increasing the probability of collision, thereby generating more energy in a short period of time. It belongs to “internal frictional heat” and effectively overcomes the phenomenon of heating hysteresis. Related studies [23,24,25] have shown that microwaves not only optimize the preparation process, but also greatly improve the catalytic performance of materials. This is because the mechanism of microwave action is related to the dielectric loss mechanism of the material, which, in turn, depends on the internal defects of the material. The mutual coupling between the microwave field and dielectric materials affects the crystal morphology and structural properties of the materials, thereby altering the catalytic performance of photocatalytic materials.

In this paper, a series of GO/TiO_2_ photocatalysts were successfully optimized using a facile microwave field strategy. By regulating the irradiation power and irradiation duration, the relationship between the photoelectrochemical properties of GO/TiO_2_ and photocatalytic efficiency was thoroughly explored, with Rhodamine B serving as the model pollutant. This work may provide a concise, effective, and environmentally friendly preparation and optimization strategy for photocatalyst synthesis.

## 2. Materials and Methods

### 2.1. Materials and Reagents

Chemicals of analytical grade were used, including concentrated sulfuric acid and hydrochloric acid (HCl, 37 wt.%) from Chuandong Chemical Co., Ltd. (Chongqing, China), potassium permanganate (KMnO_4_), hydrogen peroxide (H_2_O_2_, 30 wt.%), Rhodamine B (RhB), isopropanol (IPA), and ammonium oxalate (AO) from Chongqing Kelong Chemical Reagent Factory (Chongqing, China), sodium nitrate (NaNO_3_) from Chongqing Boyi Chemical Reagent Factory (Chongqing, China), titania (TiO_2_, P25) from Chengdu McCarthy chemical plant (Chengdu, China), and graphite powder (<150 μm) from Sigma-Aldrich Reagents (New York, NY, USA). All chemicals were used without further purification. Deionized water was used throughout the preparation. The obtained GO/TiO_2_ samples were stored in a desiccator to prevent deterioration.

### 2.2. Preparation of GO/TiO_2_ Photocatalyst

As depicted in Figure 1, graphite oxide (GO) was first prepared using a modified Hummers method [26], and then the microwave field was introduced to fabricate the GO/TiO_2_ photocatalysts.

To be specific, under ice bath conditions, 55 mL of concentrated sulfuric acid inside a three-port flask was added to 1.0 g of graphite powder, 1.0 g of NaNO_3_, and 1.0 g of KMnO_4_ successively and slowly, with stirring continued for 2 h. Next, 100 mL of deionized water was added, causing the solution temperature to rise to 35 °C, and stirring was maintained for another 4 h. After that, the three-port flask containing the mixture was moved to an oil bath at 90 °C and stirred for 30 min. Then, 200 mL of deionized water and 30 mL of H_2_O_2_ were slowly added to the mixed solution in sequence, while the temperature of the oil bath was raised to 98 °C, and stirring was continued for a final 30 min. After aging for exactly 12 h, the residue was washed twice with HCl (wt. 10%) and then with deionized water through repeated centrifugation at 5500 rpm until the pH of the supernatant approached 6. Last, the washed precipitate was dried at 60 °C for 24 h and ground to yield GO powder. In this way, about 55% of GO layers with hydroxyl and formyl acid groups were obtained from the stacked graphite.

As for the preparation of GO/TiO_2_, 50 mg of the as-prepared GO powder and 450 mg of commercial TiO_2_ nanopowder were mixed in a beaker with 15 mL of anhydrous ethanol and 15 mL of deionized water. Then, the mixture inside the beaker was placed in the ultrasonic cleaning machine for ultrasonic treatment for 1 h. After drying at 80 °C overnight, the compounds were separately placed into a corundum crucible and then radiated in a microwave furnace at different radiation powers (400 W, 600 W, and 800 W) for different radiation times (1 min, 5 min, and 10 min). The GT(Untreated) sample was obtained through solid-phase mixing, ultrasonication, and drying steps as mentioned above, without any microwave irradiation. These composites were washed with deionized water through repeated centrifugation at 5500 rpm and were then dried at 80 °C to obtain samples denoted as GT(P/t), where P and t represent the radiation power and radiation time, respectively.

### 2.3. Photocatalytic Performance Test

The photocatalytic activity of the as-prepared GO/TiO_2_ was tested in a quartz photoreactor, as illustrated in Figure 2. The activities of the samples were evaluated based on the degradation of RhB under a 500 W Xenon lamp. A total of 30 mg of the GO/TiO_2_ sample was added to 150 mL of RhB aqueous suspension (10 mg/L). Prior to irradiation, the suspensions were magnetically stirred in the dark for approximately 30 min to reach equilibrium adsorption at 25 °C. During irradiation, about 5 mL of the suspension was taken out every 10 min, and the suspension was centrifuged at 5500 rpm for 5 min to precipitate the GO/TiO_2_ photocatalyst. During light irradiation, cooling water was used to keep the model pollutant solution at a room temperature of 25 °C. The concentration of RhB was measured using a spectrophotometer at 553 nm. The rate of photocatalytic degradation was calculated as follows:η = (1 − C/C_0_) × 100%,(1)
where η is the rate of photocatalytic degradation, C represents the concentration and absorbance of the RhB solution at the time of t, and C_0_ represents the concentration and absorbance of the RhB solution at the initial time.

Active species trapping experiments were carried out in the presence of sacrificial agents, such as IPA and AO, to capture hydroxyl radicals (•OH) and photogenerated holes (h^+^). N_2_ was also used to remove oxygen from the system to capture superoxide radicals (•O_2_^−^). During each test, 150 mL of RhB aqueous suspension (10 mg/L) was added to 30 mg of samples and 3 mg of sacrificial agent.

### 2.4. Photocatalyst Characterizations

The X-ray diffraction (XRD) patterns were obtained using a Rigaku D/max-1200X X-ray diffractometer in the range of 5–90° with Cu Kα radiation. The field emission scanning electron microscope (SEM, JEOL JSM-7800F, JEOL Ltd., Beijing, China) and the transmission electron microscope (TEM, Titan G260–300, FEI Company, Hillsboro, OR, USA) were employed to investigate the phase and micro morphology of the GO/TiO_2_ hybrids. The specific surface areas of the GO/TiO_2_ samples were characterized by N_2_ adsorption using a fully automatic multi-station specific surface area and aperture analyzer (Quadrasorb 2MP, Quantachrome Instruments, CAL, USA). Fourier-transform infrared spectroscopy (FT-IR) was performed using a Nicolet IS50 FT-IR spectrometer (ThermoFisher Scientific, Waltham, MA, USA). Raman spectra were obtained using a fiber confocal Raman spectrometer (LabRAM HR Evolution, HORIBA Jobin Yvon, Paris, France). Additionally, the X-ray photoelectron spectroscopy (XPS) analysis was performed using a Thermo ESCALAB 250Xi spectrometer (ThermoFisher Scientific, Waltham, MA, USA).

### 2.5. Photoelectrochemical Detections

The UV-vis diffuse reflectance spectra (DRS) were obtained using a UV-3600 spectrophotometer (Shimadzu Scientific Instruments, Kyoto, Japan) to characterize the sample’s ability to absorb visible light. The instantaneous photocurrent response (i-t curves) and electrochemical impedance spectroscopy (EIS) were measured using an electrochemical system (CHI-760E, CHI Shanghai, Shanghai, China). In the system, 0.5 M Na_2_SO_4_ served as the electrolyte solution. The GO/TiO_2_ samples, platinum sheet, and saturated calomel electrode were used as the working electrode, the auxiliary electrode, and the reference electrode, respectively. To prepare the working electrode, 5 mg of photocatalysts were combined with 0.01 mL of Nafion solution (5 wt%), 0.75 mL of water, and 0.25 mL of ethanol to form a photocatalyst slurry by ultrasonic mixing for 30 min. Then, 0.1 mL of slurry was coated onto the fluorine-doped tin oxide (FTO) glass, covering an area of approximately 1 cm^2^. After low-temperature drying at 40 °C, the working electrode was ready for testing.

## 3. Results and Discussion

### 3.1. Catalytic Performance of GO/TiO_2_ Samples

The catalytic activities of the GO/TiO_2_ composite photocatalyst samples with different treatment times and microwave powers are depicted in Figure 3a. The photodegradation reaction rate of RhB followed the series: GT(600/5) > GT(400/5) > GT(800/5) > GT(600/1) > GT(600/10) > GT(Untreated) > TiO_2_. Obviously, the catalytic effects of the samples modified by microwave treatment were better than those of the untreated samples, which confirmed the advantages of microwave field processing. Specifically, throughout the whole photocatalysis, the catalytic performance of GT(600/5) was stable, achieving the highest degradation rate of 81.85% and the optimum reaction rate constant of 0.028 min^−1^, which is about four times higher than that of pure TiO_2_ nanoparticles. Furthermore, according to Figure 3b, the reaction rate constant of the composite GT(Untreated) without microwave treatment was only 0.015 min^−1^, and the *k* value was relatively lower in microwave treated samples. On the whole, 600 W microwave power and 5 min of microwave treatment time were regarded as the optimum process conditions.

### 3.2. Phase and Micro Morphology Analysis

#### 3.2.1. XRD

The phase structures of fresh TiO_2_ and microwave-irradiated GO/TiO_2_ samples were characterized by XRD, as shown in Figure 4. As depicted, for all GO/TiO_2_ composites, the peaks centered at 25.3°, 37.8°, 48.0°, 53.9°, 55.1°, 62.7°, 68.8°, 70.3°, 75.0°, and 82.7° were assigned to the (101), (004), (200), (105), (211), (204), (116), (220), (215), and (224) crystal planes of pure anatase TiO_2_ (JCPDS NO. 21–1272), indicating that anatase was the only crystalline phase and that microwave treatment did not change the main crystal structure of the as-obtained compounds. However, there was no characteristic peak of GO at about 10° in the XRD patterns, which may result from the low content of GO incorporated into the test sample. Coincidentally, no obvious characteristic peak of graphene occurred at 24.5°, possibly owing to the destruction of the original ordered accumulation of graphite and GO [27]. As reported [28], this phenomenon could also be explained by the covering of the characteristic peak of graphene by the strongest characteristic peak of anatase TiO_2_ around 25.3°.

To further investigate the effect of the microwave field, the average crystal size of TiO_2_ nanoparticles supported on GO was calculated using the Debye–Scherrer equation, as listed in Table 1. Comparing the data, it can be seen that the addition of GO to TiO_2_ enlarged the grain size (from 17.451 nm to 17.539 nm), whereas microwave treatment could refine the particles, since the doping of C atoms at the grain boundary hindered the direct contact between the crystal planes, thus inhibiting the growth of the grains [29]. The grain size did not change with microwave power ranging from 400 W to 600 W and showed a slight increase at 800 W, implying that the grain size was not sensitive to changes in microwave power. On the contrary, the grain size was greatly affected by microwave duration. Specifically, it is worth mentioning that when the microwave time was 5 min, the smallest grain diameter was observed. Of note, the variation pattern in the particle size of GO/TiO_2_ samples closely matched the photocatalytic effect.

#### 3.2.2. SEM and TEM

The morphologies of pure TiO_2_, GT(Untreated), and GT(600/5) were characterized using SEM images, as illustrated in Figure 5. Obviously, pure TiO_2_ nanoparticles in Figure 5c had tiny grain sizes but exhibited agglomeration and stacking phenomena. As shown in the composites, obvious differences were found between GT(Untreated) and GT(600/5), indicating changes in the surface morphology of the catalysts from the composite to the microwave-treated samples. Compared to the untreated sample in Figure 5a, the optimum sample in Figure 5b obviously exhibited less agglomeration of TiO_2_ particles and a relatively uniform distribution of TiO_2_ particles on GO, with abundant gullies and pores of different depths on the surface. Figure 5d presents a TEM image of GT(600/5), showcasing an even and tight dispersion of TiO_2_ nanoparticles across the GO substrate. This arrangement is advantageous for the creation of photogenerated charge carriers and increases the effective specific surface area, which is conducive to the photocatalytic degradation process.

#### 3.2.3. BET

Figure 6 presents the N_2_ adsorption–desorption isotherms of pure TiO_2_, GT(Untreated), and GT(600/5) composite samples, all of which belonged to type-IV isotherms, revealing typical characteristics of a mesoporous structure. In Figure 6a, the pore diameter distribution of the GT(Untreated) curve shows that pore size was concentrated in the 3.5–8 nm range, which is within the mesoporous range. By comparison, it can be clearly seen in Figure 6b that the number of pores and the pore size of GT(600/5) increased after microwave irradiation, probably due to the decrease in TiO_2_ grain size. As for pure TiO_2_, its pore size distribution in Figure 6c shows an uneven distribution of mesopores between 0–50, which was attributed to the partial coalescence phenomenon of nanoparticles. Using the BJH/DH method, the specific surface areas of pure TiO_2_ and the catalysts before and after microwave treatment were 47 m^2^/g, 137 m^2^/g, and 151 m^2^/g, respectively, indicating that the introduction of GO effectively expanded the specific surface area of TiO_2_, and that microwave treatment can effectively improve the specific surface area of the composite, thereby providing more active sites for the adsorption of organic pollutants and ultimately enhancing photocatalytic efficiency.

### 3.3. Chemical Structure Analysis

#### 3.3.1. FT-IR and Raman

FT-IR spectra exhibited the chemical bonds of the mixture, as shown in Figure 7a. The broad absorption peak near 3215 cm^−1^ was assigned to the stretching vibration of –OH, which mainly formed inside the graphite layer, stemming from the superposition vibration of weakly bound water and surface hydroxyl groups via ultrasonic stirring [30]. The absorption peaks at 1720 cm^−1^ and 1225 cm^−1^ corresponded to the stretching vibrations of C=O and C-O in the carboxyl group, respectively [31]. The peak at 1623 cm^−1^ was ascribed to the deformation vibration of adsorbed water molecules (O-H bending) and C=C stretching. In addition, the absorption peak at 1382 cm^−1^ corresponded to the deformation vibration of C-OH, and the peak at 1054 cm^−1^ was assigned to the stretching vibration of C-O in C-O-C [32]. Through a comparative analysis of GO and GT(Untreated) in Figure 7, the peaks at 1712 cm^−1^, 1225 cm^−1^, and 1054 cm^−1^ were unable to be observed in the latter, suggesting that GO underwent preliminarily reduction during preparation. A new absorption peak appeared at 468 cm^−1^, which corresponded to the absorption peak of Ti-O. After microwave modification, the -OH stretching vibration was significantly weakened, probably owing to the transformation of hydroxyl groups on the catalyst surface into stronger oxidative hydroxyl radicals (·OH) by microwave treatment [33,34].

Figure 7b enlarges and displays the characteristic Raman diffraction peaks belonging to graphite. The I_D_/I_G_ value was used to investigate the irregularity or order of the prepared GO affected by the microwave filed. Generally, the smaller the I_D_/I_G_ ratio, the fewer the defects and the higher the crystallization degree [35]. Obviously in Figure 7b, the I_D_/I_G_ value of the GT(Untreated) sample (1.74) increased compared to GO (1.62), whereas the value of the GT(600/5) hybrid (1.46) decreased significantly, indicating that microwave treatment could make the arrangement of C atoms in the GO layer tend toward a more stable sp^2^ orbit, so as to achieve a more ordered internal structure.

#### 3.3.2. XPS

Figure 8 displays the high resolution XPS spectra of C 1s and Ti 2p in the samples. Through the deconvolution fitting of C 1s patterns, three peaks attributed to C-C, C-O, and C=O were obtained, and the absorption intensity of C=O in every sample was quite weak, indicating a small amount of unreduced carboxyl groups in the hybrids. Based on the peak intensity area, the content of C-C in GT(600/5) rose compared to GT(Untreated), possibly due to more C atoms on GO presenting a stable sp^2^ orbit under microwave treatment [36,37]. Thus, the macroscopic layered structure of graphene was more obvious, which can be also confirmed by Raman and SEM analysis. As for the Ti 2p spectra of the composites, the Ti 2p_1/2_ and Ti 2p_3/2_ peaks of TiO_2_ were assigned to 463.8 eV and 458.1 eV, respectively. After calibration, the chemical valence state of the dual-state peak corresponded to Ti^4+^, suggesting that microwave modification had no effect on the valence state of the Ti element [38].

### 3.4. Photoelectrochemical Properties

#### 3.4.1. i-t Curves and EIS

To investigate the separation and transfer efficiency of electron-hole pairs in GO/TiO_2_ photocatalysts, the optoelectronic characterizations, including transient photocurrent response experiments and EIS tests, were conducted, as displayed in Figure 9. Compared to pure TiO_2_, the coupling with GO improved the separation efficiency of electron-hole pairs, as indicated by the better current response in Figure 9a and the smaller impedance in Figure 9b. Furthermore, for microwave-modified samples, GT(600/5) exhibited a higher photocurrent intensity than GT(Untreated), suggesting quicker separation and transfer efficiency of photogenerated carriers. A similar trend was observed in the EIS test in Figure 9b, where the GT(600/5) exhibited a smaller arc radius of the resistance signal, implying reduced electron-transfer resistance due to the microwave field.

#### 3.4.2. UV-Vis DRS and KM Plots

The optical absorbance abilities of pure TiO_2_ and as-obtained GO/TiO_2_ samples were examined using UV-Vis diffuse reflectance spectroscopy (DRS), as shown in Figure 10a. As observed from the DRS spectra, the absorption wavelength of the catalyst series was expanded to more than 400 nm (visible light region) compared to pure TiO_2_, exhibiting a broader response spectrum range and excellent visible light activity. In addition, a red shift was observed in the absorption spectra of the samples after microwave treatment, indicating that longer wavelengths of light were being absorbed. In particular, the GT(600/5) sample with the best photocatalytic effect, had the maximum light response wavelength at 510 nm. The corresponding bandgap energy value was calculated using the Kubelka–Munk (KM) equation [39], as depicted in Figure 10b. Obviously, the bandgap energies of the samples after microwave treatment narrowed to different degrees, compared to the untreated composite GT(Untreated) (3.14 eV) and pure TiO_2_ (3.22 eV). Of note, the narrowing trend of the bandgap in the GO/TiO_2_ samples was consistent with both the photoresponse in UV-Vis DRS and the photocatalytic performance. In particular, the bandgap of the GT(600/5) sample was calculated to be 2.90 eV, which exhibited the best photocatalytic effect, aligning with the previous detection results. In summary, DRS detection revealed that microwave treatment can further broaden the response spectrum range and narrow the bandgap energy of the hybrid photocatalyst, thereby improving the light utilization efficiency of the catalysts and promoting the catalytic effect.

### 3.5. Photocatalytic Mechanism

To investigate the photocatalytic mechanism of GO/TiO_2_ catalysts, radical trapping experiments were conducted, as shown in Figure 11. AO, N_2_, and IPA served as radical scavengers to capture h^+^, •O_2_^−^, and •OH, respectively. Apparently, for GT(600/5), in the presence of AO/IPA, the degradation of RhB significantly declined to 51.52% and 56.41%, respectively, suggesting the important role of holes and hydroxyl radicals. Of note, IPA has a greater impact on the scavenging of •OH in GT(600/5) than in GT(Untreated), implying that more hydroxyl radicals exist in GT(600/5) due to microwave treatment.

After microwave treatment, most of the oxygen-containing functional groups at the edges of oxidized graphene were reduced, and the structure tended to become highly ordered graphene. Therefore, the optimization effect of microwave treatment on GT(600/5) catalysts is reflected in two aspects, as illustrated in Figure 12.
(1)One aspect is that the refinement of TiO_2_ grain size and the increase in the specific surface area of the GO/TiO_2_ material enhanced the adsorption of dye pollutants. The introduction of the microwave field can directly act on the C atoms in GO, freeing them from the covalent bonds in GO and diffusing into the lattice of TiO_2_ as impurity atoms or interstitial atoms. New defects re thus formed, which more effectively capture electrons and holes, ultimately reducing the recombination rate of photogenerated carriers.(2)On the other hand, the narrowing of the bandgap width of GT(600/5) and the conversion of surface hydroxyl groups into more oxidative hydroxyl radicals (•OH) directly promotes the photocatalytic oxidation process of RhB into CO_2_, H_2_O, and other small molecule mineralized materials.

Thus, the plausible reaction pathway can be described as the following reaction equations:GO/TiO_2_ + *hv* → h^+^ + e^−^
(2)
h^+^ + e^−^ → heat (3)
h^+^ + Defects → Defects (h^+^) (4)
e^−^ + Defects → Defects (e^−^) (5)
h^+^ + H_2_O → •OH + H^+^
(6)
Defects (h^+^) + H_2_O → •OH + H^+^
(7)
h^+^ + OH^−^ → •OH (8)
Defects (h^+^) + OH^−^ → •OH (9)
e^−^ + O_2_ → ∙•O_2_^−^
(10)
Defects (e^−^) + O_2_ → ∙•O_2_^−^
(11)
•OH/h^+^/•O_2_^−^ + RhB → ∙CO_2_ + H_2_O + small molecules (12)

## 4. Conclusions

In this work, a series of GO/TiO_2_ photocatalysts were successfully modified by microwave field and exhibited more excellent photodegradation performance of RhB compared to other homologous GO/TiO_2_ catalysts (Table 2). The results illustrate that the introduction of the microwave field directly affects the C atoms in GO, freeing them from the covalent bonds in GO and diffusing into the lattice of TiO_2_ in the form of impurity atoms or interstitial atoms. New defects were thus formed, which more effectively capture electrons and holes, ultimately reducing the recombination rate of photogenerated carriers. Secondly, the microwave field can break the hydrogen bonds of water molecules in GO layers, release more active sites, and cause hydroxyl groups to dissociate to form hydroxyl radicals with strong oxidation ability, thereby increasing photocatalytic efficiency. Therein, GT(600/5) performed the most excellent photooxidation rate of RhB under visible light compared to other homologous samples, owing to the minimum grain size (16.914 nm), enlarged specific surface area (151 m^2^/g), narrowest bandgap width (2.90 eV), and stronger oxidized hydroxyl radicals (•OH). As determined by i-t curves and EIS tests, microwave irradiation at 600 W and 5 min on GO/TiO_2_ photocatalyst promoted interfacial charge transfer and suppressed charge recombination. For the sake of green environmental protection, this work may provide a concise, effective, and environmentally friendly preparation and optimization strategy for photocatalyst synthesis.

## Figures and Tables

**Figure 1 nanomaterials-14-01912-f001:**
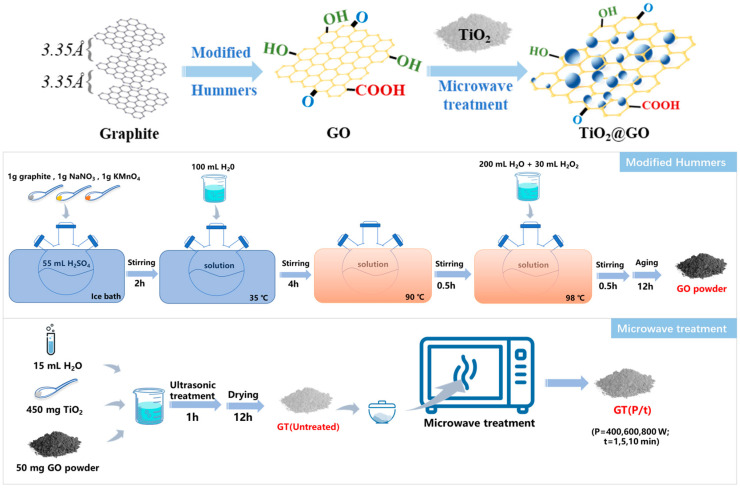
Preparation diagram of microwave-modified GO/TiO_2_ hybrids.

**Figure 2 nanomaterials-14-01912-f002:**
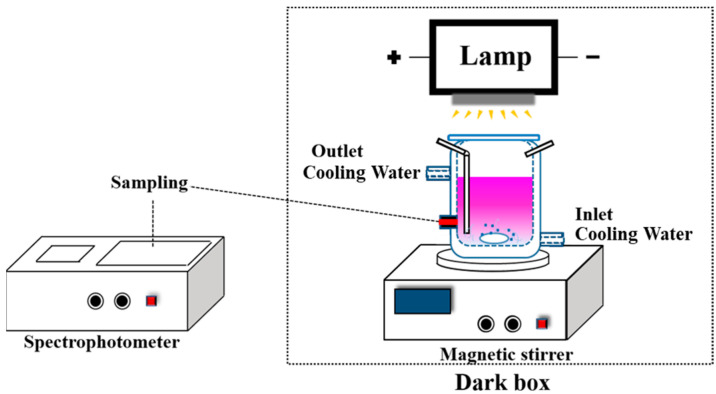
Experimental device for the photocatalytic degradation of RhB.

**Figure 3 nanomaterials-14-01912-f003:**
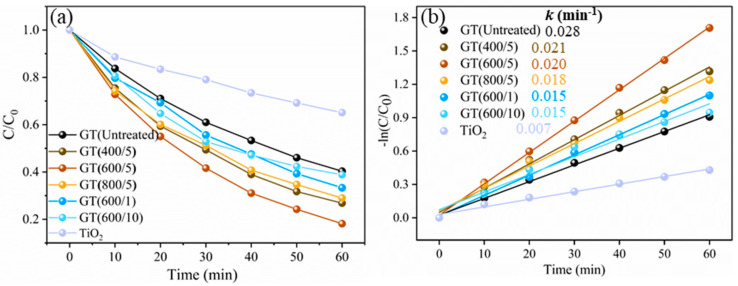
Photocatalytic test of GO/TiO_2_ photocatalysts. (**a**) Photocatalytic degradation efficiency; (**b**) the corresponding pseudo-first-order kinetics.

**Figure 4 nanomaterials-14-01912-f004:**
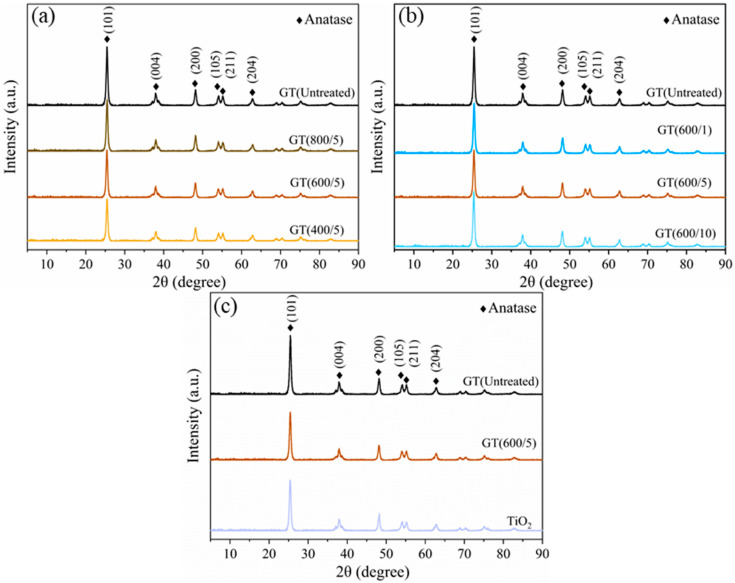
XRD characterization of GO/TiO_2_ photocatalysts. (**a**) Samples under different microwave powers; (**b**) samples under different microwave times; (**c**) comparison of the optimal GT(600/5), GT(Untreated), and pure TiO_2_.

**Figure 5 nanomaterials-14-01912-f005:**
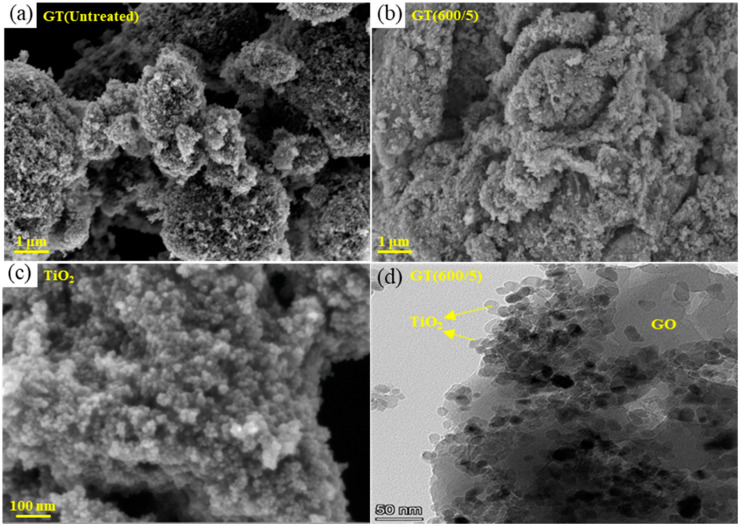
SEM images of GO/TiO_2_ photocatalysts. (**a**) GT(Untreated); (**b**) GT(600/5); (**c**) TiO_2_; (**d**) TEM image of GT(600/5).

**Figure 6 nanomaterials-14-01912-f006:**
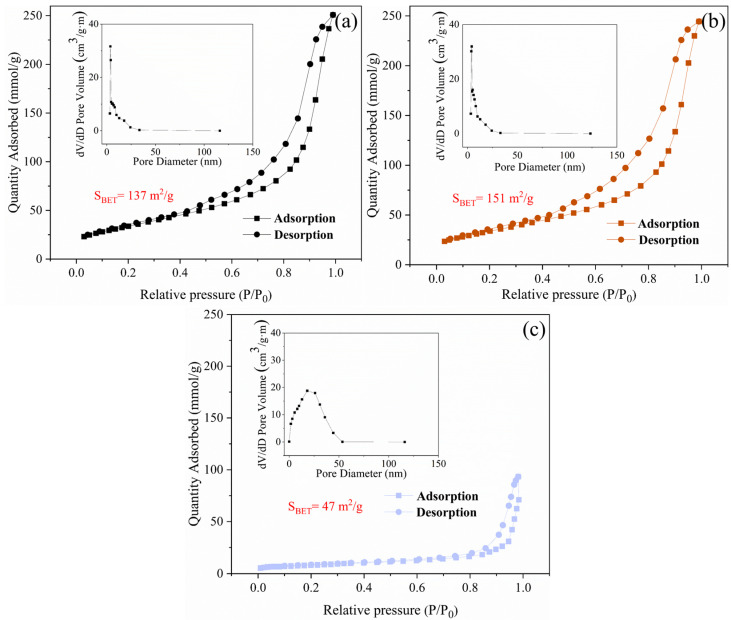
N_2_ adsorption–desorption isotherms of TiO_2_ and GO/TiO_2_ photocatalysts, with inserts of aperture distribution. (**a**) GT(Untreated); (**b**) GT(600/5); (**c**) TiO_2_.

**Figure 7 nanomaterials-14-01912-f007:**
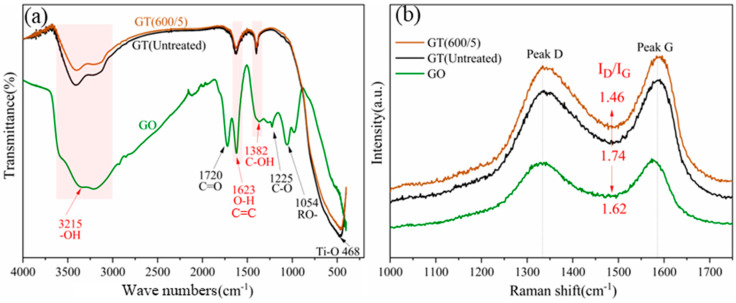
Characterizations of GO/TiO_2_ photocatalysts. (**a**) FT-IR spectra; (**b**) Raman patterns.

**Figure 8 nanomaterials-14-01912-f008:**
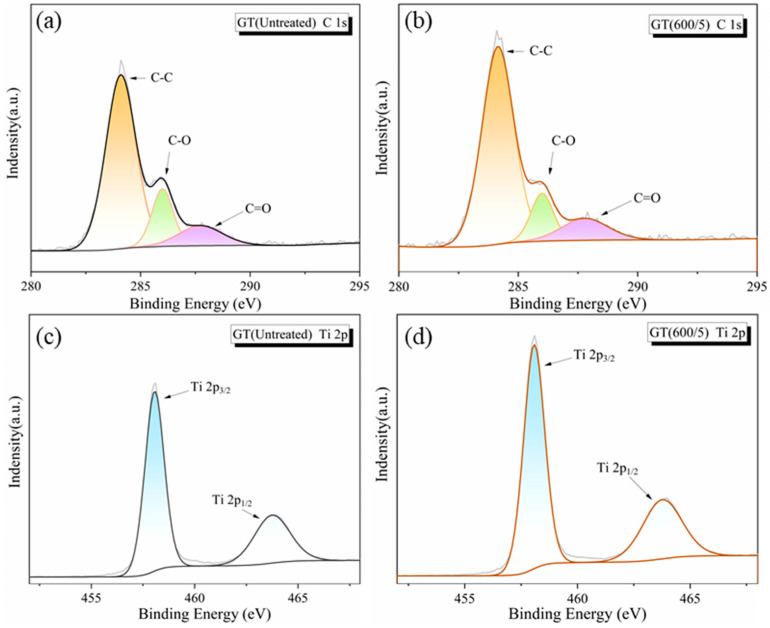
XPS spectra of GO/TiO_2_ photocatalysts. (**a**) C 1s of GT(Untreated); (**b**) C 1s of GT(600/5); (**c**) Ti 2p of GT(Untreated); (**d**) Ti 2p of GT(600/5).

**Figure 9 nanomaterials-14-01912-f009:**
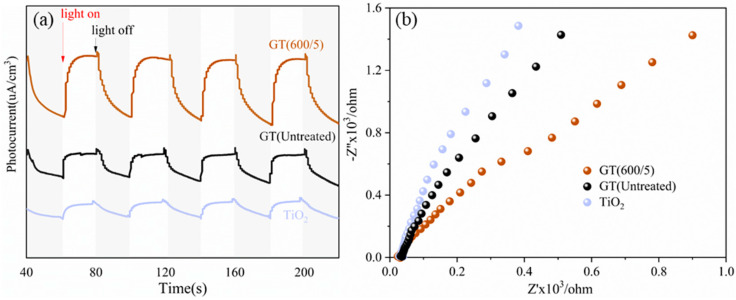
Optoelectronic characterizations of TiO_2_ and GO/TiO_2_ photocatalysts. (**a**) Transient photocurrent response (i-t curves); (**b**) EIS curves.

**Figure 10 nanomaterials-14-01912-f010:**
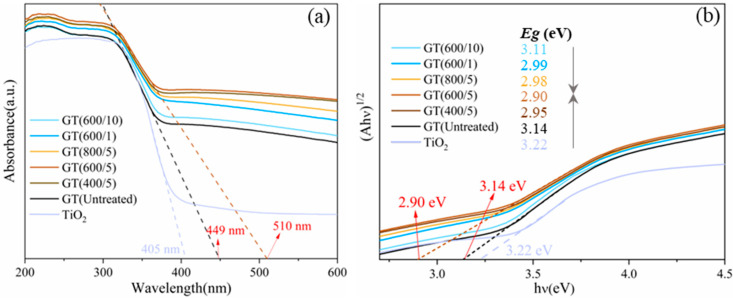
(**a**) The UV-Vis DRS spectra of TiO_2_ and GO/TiO_2_ photocatalysts; (**b**) the corresponding Kubelka–Munk plots and bandgaps of TiO_2_ and GO/TiO_2_ photocatalysts.

**Figure 11 nanomaterials-14-01912-f011:**
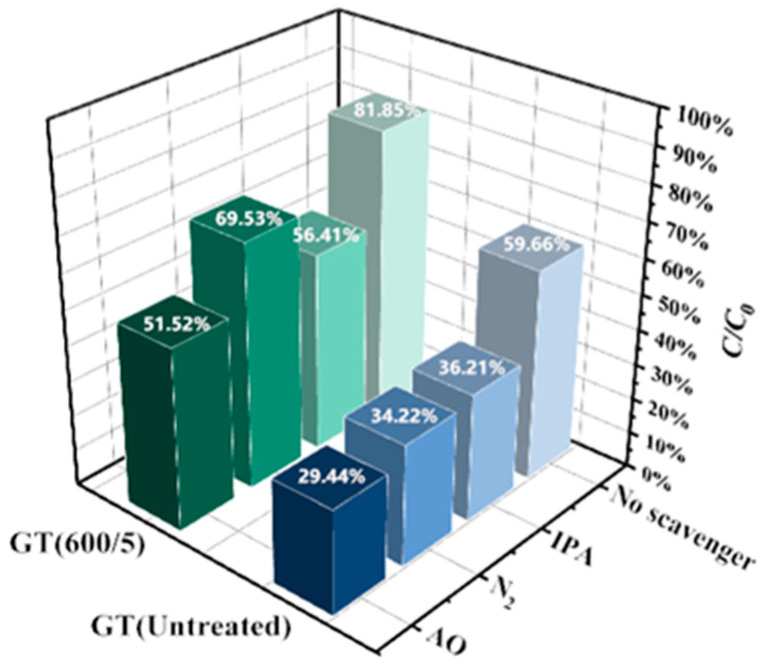
Active species trapping experiments of GT(Untreated) and GT(600/5).

**Figure 12 nanomaterials-14-01912-f012:**
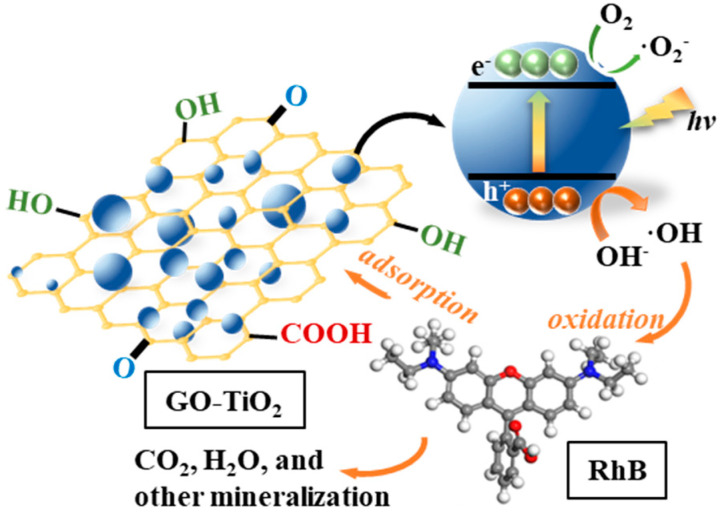
Possible photocatalytic mechanism of RhB degradation by GO/TiO_2_ photocatalysts.

**Table 1 nanomaterials-14-01912-t001:** Calculation of GO/TiO_2_ samples based on XRD pattern.

Samples	D_(101)_ (nm) ^1^	D’ (nm) ^2^
GT(400/5)	16.948	176
GT(600/5)	16.914	172
GT(800/5)	17.130	185
GT(600/1)	17.203	187
GT(600/10)	17.311	189
GT(Untreated)	17.539	191
TiO_2_	17.451	190

^1^ D_(101)_—Grain diameter on the crystal plane of (101). ^2^ D’—the average grain diameter on the crystal plane corresponding to each characteristic peak within the range of 0–90°.

**Table 2 nanomaterials-14-01912-t002:** Current works on RhB photodegradation using homologous GO/TiO_2_ catalysts.

Sample	Method	Light Source	Pollutant	Degradation Rate (%)	Ref.
TiO_2_/ZnO/rGO	Solvothermal	UV-light	RhB (5 mg/L)	100%, 90 min	[40]
TiO_2_/GO/H_2_O_2_	Sol-gel	UV-light	RhB (10 ppm)	100%, 80 min	[41]
GO/TiO_2_/polycalix	One-pot	Solar	RhB (5 ppm)	70%, 120 min	[42]
1% TiO_2_/GO	Microwave	Xe-Lamp	RhB (8 ppm)	30%, 120 min	[43]
GT(600/5)	Microwave	Visible	RhB (10 mg/L)	81.5%, 60 min	This work

## Data Availability

Data underlying the results presented in this paper are not publicly available at this time, but may be obtained from the authors upon reasonable request.

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
