# Peer review of "Microwave-Field-Optimized GO/TiO2 Nanomaterials for Enhanced Interfacial Charge Transfer in Photocatalysis"

_nanomaterials, 2024, doi:10.3390/nano14231912_

Round 1
Reviewer 1 Report
Comments and Suggestions for Authors
The paper describes the synthesis, characterization and photocatalytic properties of GO/TiO2 prepared by MW method. The topic is interesting, and the photocatalytic mechanism study is appreciable. The physicochemical characterization on photocatalysts was carried out using a wide range of instrumentation. Nonetheless, it is incomplete because all the samples, including TiO2 and GO alone, should be subjected to all techniques. The authors simply compare the best sample GT(600/5) with the untreated one, but it could be interesting to verify the behavioral trends for all MW synthesis conditions. In section 3.4.2. UV- Vis DRS and KM plots, although all the data was reported, it was only discussed the comparison between GT(Untreated) and GT(600/5). The trend among the other samples appears strange, the authors should provide some explanations. Moreover, the instrumental method and the relative samples’ preparation should be expanded and improved. It is unclear how the GT(Untreated) sample was prepared. The conclusion should be expanded and improved.
The manuscript could become acceptable after major revision.
Author Response
Comments 1: The physicochemical characterization on photocatalysts was carried out using a wide range of instrumentation. Nonetheless, it is incomplete because all the samples, including TiO2 and GO alone, should be subjected to all techniques. The authors simply compare the best sample GT(600/5) with the untreated one, but it could be interesting to verify the behavioral trends for all MW synthesis conditions.
Response 1: Thank you for pointing this out. Your suggestion is very meaningful and provides important guidance for our research work. However, we have to explain that the content of the article focuses more on the effect of microwave modification on photo generated charge carriers in GO/TiO2 composites rather than on monomers. Through comparative testing of GT (600/5) and GT (Untreated) samples, the maximum change produced under the optimal microwave modification conditions was intuitively expressed. At the same time, in order to correspond to the catalytic activity of the series of samples, we measured the photoresponsivity and corresponding bandgap width of all samples, and obtained consistent trends. Thank you again for your comments and we will actively adopt it in future research.
Comments 2: In section 3.4.2. UV- Vis DRS and KM plots, although all the data was reported, it was only discussed the comparison between GT(Untreated) and GT(600/5). The trend among the other samples appears strange, the authors should provide some explanations.
Response 2: Thanks very much for your reminder. Exactly, the narrowing trend of bandgap in GO/TiO2 samples was consistent with both the photoresponse in UV-Vis DRS and the photocatalytic performance. We have added relevant statements in page 10, line 307-308.
Comments 3: Moreover, the instrumental method and the relative samples’ preparation should be expanded and improved. It is unclear how the GT(Untreated) sample was prepared.
Response 3: Thanks for your meticulous inspection. Contents about the instrumental method and the relative samples’ preparation have been expanded in detail and the step for GT(Untreated) sample preparation was also added in the revised version (seen in page 3, line 105-106, Fig. 1).
Comments 4: The conclusion should be expanded and improved.
Response 4: Thanks for your valuable suggestions for this article. The conclusion have been expanded with reasonable content regarding the mechanism section (seen in page 12, line 364-373).
Reviewer 2 Report
Comments and Suggestions for Authors
Based on my review of the manuscript titled “Microwave Field Optimized GO/TiO2 Nanomaterials for Enhanced Interfacial Charge Transfer in Photocatalysis”, the present manuscript explores the use of nanomaterials based on TiOâ‚‚ and graphene oxide (GO) for photocatalysis, which is a relevant topic in current research due to its potential for pollutant degradation. Clear and Well-Defined Methodology: The methodology, including the use of microwave field treatment and characterizations (XRD, SEM, BET, FT-IR, Raman, XPS), is described in detail, providing a solid basis for reproducibility. This article presents a complete analysis of the photoelectrochemical properties and charge transfer mechanisms using tests such as photocurrent response and EIS to evaluate the efficiency of electron-hole pair separation. Detailed experimental data are presented to support the conclusions, such as the increase in surface area and the reduction in bandgap for optimized samples.
On the other hand, the authors improved the manuscript as follows:
- The discussion of the results would be more direct and concise. At times, the analysis seems repetitive and could be simplified to avoid redundancies and improve the reading flow.
- It would be interesting to conduct a direct comparison with other similar studies to highlight how the methods and results compare to existing solutions, which would help emphasize the innovation of the work.
- Although this mechanistic proposal is presented, a more in-depth description of the role of hydroxyl radicals and other reactive species would be beneficial for understanding the contribution of each experimental factor.
- Including a flow diagram or schematic that visually demonstrates the proposed mechanism for increasing the catalytic efficiency could facilitate understanding, especially for readers who are not specialists.
- The abstract does not define an EIS test. It is recommended that acronyms be defined when they are mentioned for the first time in the text.
- Specific surface area values should not be used in decimal places.
- The abstract should include more quantitative photocatalysis test results.
- Between lines 65 and 72, the authors provide conclusive sentences. These types of comments should not appear in an introduction. I strongly recommend that they be removed.
- In section 2.3 Photocatalytic performance test, was the pH of the reaction mixture (Rhodamine solution and photocatalyst) adjusted?
- Between lines 126 and 129, the authors state that they performed active species trapping experiments. I assume they are referring to the Scavereng Test, which is well known in the scientific community. Therefore, I would ask you to describe these experiments better by adding the amount of rhodamine solution, the number of species trapped, and their respective concentrations.
Comments on the Quality of English LanguageOverall, with a few revisions in structure, clarity, and consistency, the English of the article was improved, making it even more comprehensible to the scientific community.
Author Response
Comments 1: The discussion of the results would be more direct and concise. At times, the analysis seems repetitive and could be simplified to avoid redundancies and improve the reading flow.
Response 1: Thank you for pointing this out. We have reorganized the language for partial discussion to make it concise and appropriately expressed.
Comments 2: It would be interesting to conduct a direct comparison with other similar studies to highlight how the methods and results compare to existing solutions, which would help emphasize the innovation of the work.
Response 2: We agree with this comment. Therefore, we added a comparison table with existing literature research at the end of the discussion section of the article.
Comments 3: Although this mechanistic proposal is presented, a more in-depth description of the role of hydroxyl radicals and other reactive species would be beneficial for understanding the contribution of each experimental factor.
Response 3: Thanks for your valuable suggestion. The plausible reaction pathway has been described as the detailed reaction equations of reactive species added in the Chapter 3.5.
Comments 4: Including a flow diagram or schematic that visually demonstrates the proposed mechanism for increasing the catalytic efficiency could facilitate understanding, especially for readers who are not specialists.
Response 4: Thanks for your suggestion. We have reorganized the language expression of the mechanism section and added intuitive reaction equations to help non-professionals understand.
Comments 5: The abstract does not define an EIS test. It is recommended that acronyms be defined when they are mentioned for the first time in the text.
Response 5: We have replace the abbreviation of EIS in the abstract with "electrochemical impedance spectroscop (EIS)".
Comments 6: Specific surface area values should not be used in decimal places.
Response 6: Thanks for your careful observation. We have corrected the improper use of data formats.
Comments 7: The abstract should include more quantitative photocatalysis test results.
Response 7: We have added necessary photocatalytic performance and light response wavelength in the abstract.
Comments 8: Between lines 65 and 72, the authors provide conclusive sentences. These types of comments should not appear in an introduction. I strongly recommend that they be removed.
Response 8: We have removed inappropriate content in the introduction as you mentioned.
Comments 9: In section 2.3 Photocatalytic performance test, was the pH of the reaction mixture (Rhodamine solution and photocatalyst) adjusted?
Response 9: The pH of the reaction mixture was not adjusted during the test.
Comments 10: Between lines 126 and 129, the authors state that they performed active species trapping experiments. I assume they are referring to the Scavereng Test, which is well known in the scientific community. Therefore, I would ask you to describe these experiments better by adding the amount of rhodamine solution, the number of species trapped, and their respective concentrations.
Response 10: Thanks for your meticulous inspection. We have added related information at the end of Chapter 2.3.
Reviewer 3 Report
Comments and Suggestions for Authors
In this article, the authors discuss the microwave synthesis of GO/TiO2 composites at different irradiation power and duration. Furthermore, the structure, morphology, composition, photochemical and photoelectrochemical activity of the obtained composites are examined in detail. The manuscript is written clearly and coherently, mostly arguing, justifying, and discussing the results obtained during the experiments. The work contains some practical aspects, which were worth to be investigated.
However, I think some things need to be added to the manuscript to show the uniqueness/superiority of the obtained GO/TiO2 composites compared to the "classical" TiO2 photocatalyst. After these essential additions and some revisions, the manuscript would be eligible for publication in the journal “Nanomaterials”.
My suggestions, questions and comments that would improve the quality of the manuscript are as follows:
- In the Abstract, the abbreviation RhB should not be used without its full meaning;
- In Chapter "1. Introduction” (line 48), please correct the word “modificatrion”;
- In Chapter "2.1. Materials and reagents”, information about the particle size or mesh of TiO2 and graphite powders should be provided. Please, indicate the supplier or producer of graphite powder also;
- In Chapter "2.2. Preparation of GO/TiO2 photocatalyst” the yield of GO obtained by the described method (approximately, in percents) should be indicated;
- I recommend changing the name of Chapter 2.5. to “Photoelectrochemical activity test”. Here, the preparation of the working electrode should also be described in detail (the material of the substrate, applied method of deposition of the photochemically active substances under study on this substrate, additional materials used to form the coating, etc.);
- The statement about the no agglomeration of TiO2 particles and the uniform distribution of TiO2 particles on graphene oxide (lines 208-209) are incorrect and completely unsupported by the provided SEM images. These images do not show which material is where. Additional SEM pictures of individual TiO2 particles and obtained GO particles, as well as elemental mapping images obtained during EDS analysis would allow a clearer understanding of the real distribution/location of TiO2 and GO particles in the composite;
- To prove the exceptional photochemical and photoelectrochemical efficiency of the obtained composites, the results of all the tests performed with the composites (untreated and treated by microwaves) should be accompanied by the results of analogous tests for pure TiO2, i.e. Figures 3, 4, 6, 7a), 8, 9 and 10 should be supplemented with the corresponding curves for pure TiO2. The text should also discuss the properties of the obtained composites in comparison with pure TiO2;
- In Chapter "3.3.2. XPS”, it is stated that "the content of C-C in GT(600/5) rose compared to GT(Untreated), may due to the return of sp3 hybridized C atoms on GO to a more stable sp2 orbit under microwave". Such a statement raises reasonable doubts because the values of element concentrations determined by the XPS method are not provided, and also the changes in the hybridization of C atoms are not reflected in the change/shift of the Binding Energy (Figure 8). Please see https://doi.org/10.1016/j.apsusc.2018.04.269.
Author Response
Reviewer #3:
Comments 1: In the Abstract, the abbreviation RhB should not be used without its full meaning;
Response 1: Thanks for your meticulous inspection. We have replace the abbreviation of RhB in the abstract with "Rhodamine B".
Comments 2: In Chapter "1. Introduction” (line 48), please correct the word “modificatrion”;
Response 2: We are sorry for the misunderstanding expression and have corrected the mentioned error in the revised manuscript.
Comments 3: In Chapter "2.1. Materials and reagents”, information about the particle size or mesh of TiO2 and graphite powders should be provided. Please, indicate the supplier or producer of graphite powder also;
Response 3: The relevant information as you mentioned has been supplemented in the corresponding position in Chapter 2.1.
Comments 4: In Chapter "2.2. Preparation of GO/TiO2 photocatalyst” the yield of GO obtained by the described method (approximately, in percents) should be indicated;
Response 4: About 55% GO layers with hydroxyl and formyl acid groups were obtained from stacked graphite through the modified Hummers method. The relevant information has been added in Chapter 2.2.
Comments 5: I recommend changing the name of Chapter 2.5. to “Photoelectrochemical activity test”. Here, the preparation of the working electrode should also be described in detail (the material of the substrate, applied method of deposition of the photochemically active substances under study on this substrate, additional materials used to form the coating, etc.);
Response 5: Thanks for your valuable suggestions. Detailed operation for working electrode has been added to this testing section.
Comments 6: The statement about the no agglomeration of TiO2 particles and the uniform distribution of TiO2 particles on graphene oxide (lines 208-209) are incorrect and completely unsupported by the provided SEM images. These images do not show which material is where. Additional SEM pictures of individual TiO2 particles and obtained GO particles, as well as elemental mapping images obtained during EDS analysis would allow a clearer understanding of the real distribution/location of TiO2 and GO particles in the composite;
Response 6: Agree. Thus, we have reorganized language expression and added the TEM image of GT(600/5) in Fig. 3(c) to observe GO and TiO2 of the sample and the relevant discussion can be found in page 6, line 216-221 .
Comments 7: To prove the exceptional photochemical and photoelectrochemical efficiency of the obtained composites, the results of all the tests performed with the composites (untreated and treated by microwaves) should be accompanied by the results of analogous tests for pure TiO2, i.e. Figures 3, 4, 6, 7a), 8, 9 and 10 should be supplemented with the corresponding curves for pure TiO2. The text should also discuss the properties of the obtained composites in comparison with pure TiO2;
Response 7: Thank you for pointing this out. Your suggestion is very meaningful and provides important guidance for our research work. However, we have to explain that the content of the article focuses more on the effect of microwave modification on photo generated charge carriers in GO/TiO2 composites rather than on monomers. Through comparative testing of GT (600/5) and GT (Untreated) samples, the maximum change produced under the optimal microwave modification conditions was intuitively expressed. At the same time, in order to correspond to the catalytic activity of the series of samples, we measured the photoresponsivity and corresponding bandgap width of all samples, and obtained consistent trends. Thank you again for your comments and we will actively adopt it in future research.
Comments 8: In Chapter "3.3.2. XPS”, it is stated that "the content of C-C in GT(600/5) rose compared to GT(Untreated), may due to the return of sp3 hybridized C atoms on GO to a more stable sp2 orbit under microwave". Such a statement raises reasonable doubts because the values of element concentrations determined by the XPS method are not provided, and also the changes in the hybridization of C atoms are not reflected in the change/shift of the Binding Energy (Figure 8). Please see https://doi.org/10.1016/j.apsusc.2018.04.269.
Response 8: Thanks for your valuable suggestions for this article. We have reorganized language expression and cited the aforementioned literature.
Reviewer 4 Report
Comments and Suggestions for Authors
Dear Editor,
Thanks for inviting me for reviewing the manuscript entitled Microwave Field Optimized GO/TiO2 Nanomaterials for Enhanced Interfacial Charge Transfer in Photocatalysis authored by Xu Duan, Weizao Liu, and Jing Guo.
The authors present a novel approach by using microwave irradiation to modify GO/TiO2 nanocomposites, which significantly enhances photocatalytic efficiency. This is a well-conceived and interesting study, as microwave treatment indeed appears to reduce electron-hole recombination. However, there are still some issues that need to be addressed before its publication in Nanomaterials.
1) The photocatalysis of GO combined with TiO2 has been many reported by others, the authors should enhance the readability of the introduction with more discussion.
2) TEM images of GO/TiO2 are suggested to be provided so the quality and morphology of GO and TiO2 can be viewed.
3) It would be better if the authors could further discuss the mechanisms by which microwave treatment influences the specific properties of GO and TiO2 in terms of crystallinity and defect formation.
4) In the reference, please check [9] and [32] for the number subscription and page number.
Author Response
Comments 1: The photocatalysis of GO combined with TiO2 has been many reported by others, the authors should enhance the readability of the introduction with more discussion.
Response 1: Thank you for pointing this out. We agree with this comment. Therefore, we have added some discussion in the introduction for the wider readership. This change can be found in page 2, line 46-50 in the revised manuscript.
Comments 2: TEM images of GO/TiO2 are suggested to be provided so the quality and morphology of GO and TiO2 can be viewed.
Response 2: Agree. Thus, we have added a TEM image of GT(600/5) in Fig. 3(c) to observe GO and TiO2 of the sample and the relevant discussion can be found in page 6, line 216-221 .
Comments 3: It would be better if the authors could further discuss the mechanisms by which microwave treatment influences the specific properties of GO and TiO2 in terms of crystallinity and defect formation.
Response 3: Thanks for your valuable suggestions for this point. We have reorganized the language expression of the mechanism section and added intuitive reaction equations related to defects participation in the photocatalytic reactions.
Comments 4: In the reference, please check [9] and [32] for the number subscription and page number.
Response 4: Thank you for your detailed reminder. We have checked the whole references and corrected missing page numbers of [9], [10], [28] and [32] in the original text.
Round 2
Reviewer 1 Report
Comments and Suggestions for Authors
The authors replied only partially to the comments. In the specific, the characterization on TiO2 and GO alone is fundamental. Although the focus was the effect of microwave, the comparison with starting materials (TiO2 and GO) is important.
Author Response
Comments 1:The authors replied only partially to the comments. In the specific, the characterization on TiO2 and GO alone is fundamental. Although the focus was the effect of microwave, the comparison with starting materials (TiO2 and GO) is important.
Response 1: Thank you for pointing this out. We agreewith this comment. Therefore, we have added detection and related analysis discussion of TiO2 and GO monomers,which can be found in red from chapter '3. Results and discussion' in the revised manuscript.
Reviewer 2 Report
Comments and Suggestions for Authors
The authors have made all the recommended corrections. I therefore agree ​​​​​​​to the publication of the manuscript in the journal.
Author Response
Thank you very much for your recognition of the revised manuscript. Your expertise and constructive feedback have contributed to the enhancement of this particular manuscript.
Reviewer 3 Report
Comments and Suggestions for Authors
I thank the authors for the appropriate response to the comments and suggestions. I have no additional questions or comments.
Author Response

(The authors gave the same response as above.)

Reviewer 4 Report
Comments and Suggestions for Authors
The comments and suggestions from this reviewer have been well addressed, and I dont have further comments on it. So it can be accepted.
Author Response

(The authors gave the same response as above.)

Round 3
Reviewer 1 Report
Comments and Suggestions for Authors
The paper has been improved and it is acceptable for the publication.